

**Holocene Proxy Climate Series Should Account for the Site's Elevation, the Variable's Sensitivity to**
**Elevation History and Time-lagged  Effects: Three Examples**
David A Fisher
Department of Earth Sciences , University of Ottawa, Ottawa Ontario,  Canada
**Abstract**
When making multi-proxy reconstructions over Holocene-long periods , an argument is presented
that the elevation of the sites used and/or their elevation  history must be taken into account before
their proxy records (of temperature or precipitation) are included in the reconstruction. It is shown that
to ignore elevation, results in first order errors in the reconstruction, especially in regions under (or
close to) the degrading ice sheet .  Also it is argued that when assessing the signature of a given putative
global event  (like the 4.2 ka event) , one must allow for there being a complex  signature wrt. location,
elevation and time lagged variables. Three specific examples are used to illustrate these points.

**1.  Introduction**
As the title suggests this contribution is an argument for the necessity of including the elevation
and/or changes in elevation of the sites used in reconstructions because:
1) The variables used could be very elevation sensitive.
2) The site could undergo large changes in elevation throughout the Holocene.
3) There could be a lagged relationship between some sites widely separated, because the ocean delay
time is critical factor in their relationship.
Each of the three cases is illustrated with an example.
**2.  Stable Isotope Signatures of the Same Event at One Geographical Location,  SW  Yukon, but From**
**Sites Spread Over  5 Vertical Kilometers.**
The Mt Logan $\delta(^{18}O)$ record  (Figure 1a) has been interpreted, not as a temperature series, but as a
proxy for the strength of El Niňo,  (Fisher et al., 2008).  The Logan  $\delta(^{18}O)$ series has its largest change,  ~
6  o/oo, between  4.2 a to 4.0 ka B2k, (Walker et al.,  2012). There are many other large negative
excursions in the Logan Holocene including the one about AD 1835, which is ~ 4  o/oo deep. This AD 1835
change has been captured also at other nearby sites that have very different elevations. In the Eclipse
ice core (3000 m. a.s.l.)  the AD 1835 event showed a near zero  $\delta(^{18}O)$  shift and in the Jelly Bean Lake
record (catchment elevation 1000- 1500 m. a.s.l.) the 1835 event has a $\delta(^{18}O)$ shift of ~ 1.2  o/oo. For
records below 1000 m. a.s.l  the shift is actually in the opposite direction. So the size and even sign of
this 1835 large $\delta(^{18}O)$  shift depends strongly on the site elevation.  Figure 2 points show the size of this





AD 1835  δ($^{18}$O) shift as a function of elevation. The line with shading shows what a  δ($^{18}$O) model
produces for the AD 1835 shift. Elevation is clearly  a critical variable for interpreting proxy records when
stable isotopes are the variable measured. Putting the Logan  δ($^{18}$O) record in a regional scale statistical
analysis (Fisher, 2002) and finding  principal components  of temperature sensitive paleo-records,
makes it stand out as being out phase with regional  temperature, (Fisher, 2002).  Figure 3 shows the 1$^{st}$
principal component of a 51 multi-proxy suite, most of which, are temperature-like  (circles) and 7
regional measured temperature series (squares)  all spanning from AD 1761 to AD 1970.  This time
period covers the main recovery from the Little Ice Age (LIA) and includes AD 1835. Logan δ($^{18}$O) , stands
out as being out of phase as does its nearest neighbour. The large majority of these series record the
recovery from the LIA. But the  Logan  δ($^{18}$O) site, being so high and adjacent to the Eastern Pacific,  does
not record local temperature but rather the moisture source distribution through time, (Fisher et al.,
45    2004, 2008).


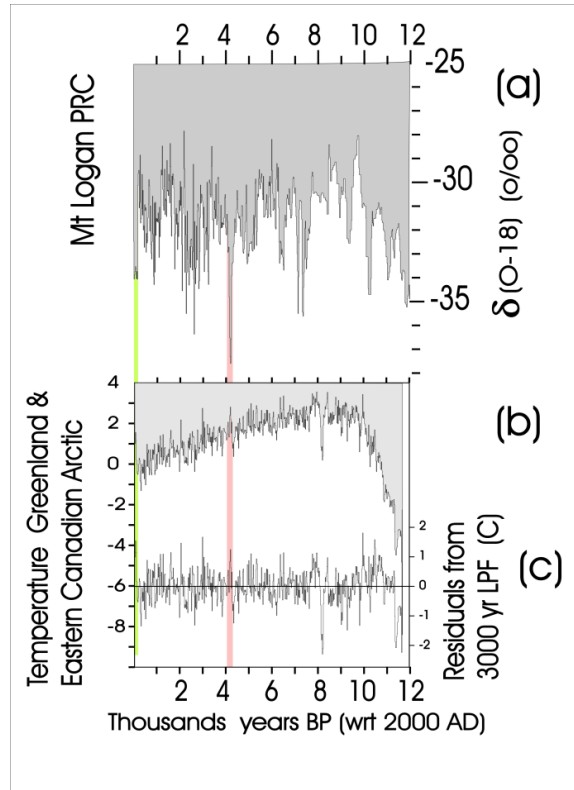

**Figure 1**; a) Mt Logan δ($^{18}$O)  (Yukon) record (Fisher ,2011) shows a very strong 4.2 ka event,
shaded red and the shift in AD 1835, shaded green.  b) Elevation corrected  δ($^{18}$O)-based
temperature record from Renland (Greenland) plus Agassiz (Ellesmere Island) ice cores, (Vinther
et al.,2009). c) Residuals from 3000 year trend line through b.




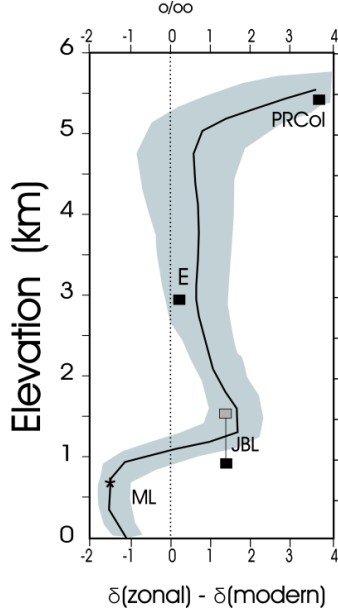


**Figure 2** Size of the $\delta(^{18}O)$ shifts at ~ AD 1835, in various records from the SW Yukon versus elevation,
(Fisher et al., 2008). **PRCol** is the Logan ice core, **E** is Eclipse ice core, **JBL** Jelly Bean lake (Anderson et
al., 2005) and **ML** is Marcella Lake (Anderson et al., 2007). The line gives a model prediction of what
happened to $\delta(^{18}O)$, when at AD 1835 there was a shift from moisture sources; from zonal to meridional
and the shading is the range of model outputs over a range of input assumptions (Fisher et al., 2004) .





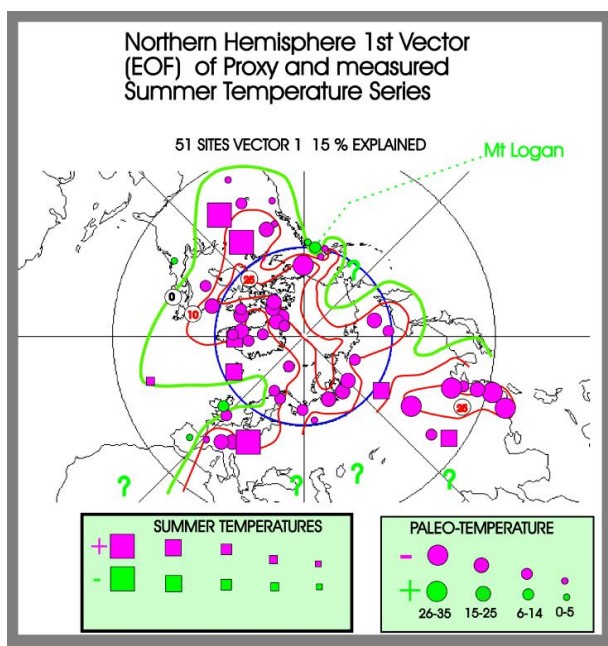

60

**Figure 3** The 1st principal component of 51 proxy sites (circles)most of which are temperature-like and 7
regional measured temperature sites (squares) all spanning the years AD 1761 to AD 1970. The size of
the circle at a given site is proportional its "suite-covariance" and the colour declares the sign. All red
sites are in phase with each other. The total variance explained for the 51 proxy sites by the 1st PC is
15%, which is significant at the 95% level, see (Fisher, 2002). The Mt Logan site, 5400 m. a.s.l., is
indicated.

67

**3. Example of the Need to Make Corrections to Holocene-long Stable Isotope (and melt) Records
from sites on (or Near), a Changing Ice Sheet.**

Looking at the Holocene δ($^{18}$O) (or melt) series from the many ice cores from Greenland and from NE
Arctic Canada one sees that, while many of the high frequency details are the same over a wide
geographical extent (see Figure 4), the general trends can be quite different and the maximum warmth,
(Figure 4b) between sites occurs at different ages. In the Renland and Agassiz cores the early Holocene
is clearly the warmest part of the Holocene, unlike the deep cores on the Ice Sheet that have different
generalized δ($^{18}$O) histories, Figure 4b. The reasons for the differences in general slope and age of
maximum warmth has been shown to be largely due to the original series having been affected by their
Holocene history of ice thickness and bedrock depression. When model calculated corrections are made
on the (bracketing site's) Renland and Agassiz elevation histories, the climate history across this region
emerges and the differences between the deep ice cores can be attributed plausibly to different ice
thickness and bed rock depression histories throughout the Holocene, (Vinther et al., 2009; Lecavalier
et al., 2017). Not allowing for the history of the 3$^{rd}$ dimension through time-lagged elevation responses



has caused some investigators, who examined the geographical patterns of Holocene proxy temperature
based on uncorrected  δ($^{18}$O), to conclude the Holocene warmth over the Greenland ice sheet was not
always in the early Holocene.  So ignoring the additional dimension of elevation through  time results in
erroneous conclusions, (Briner et al., 2016).

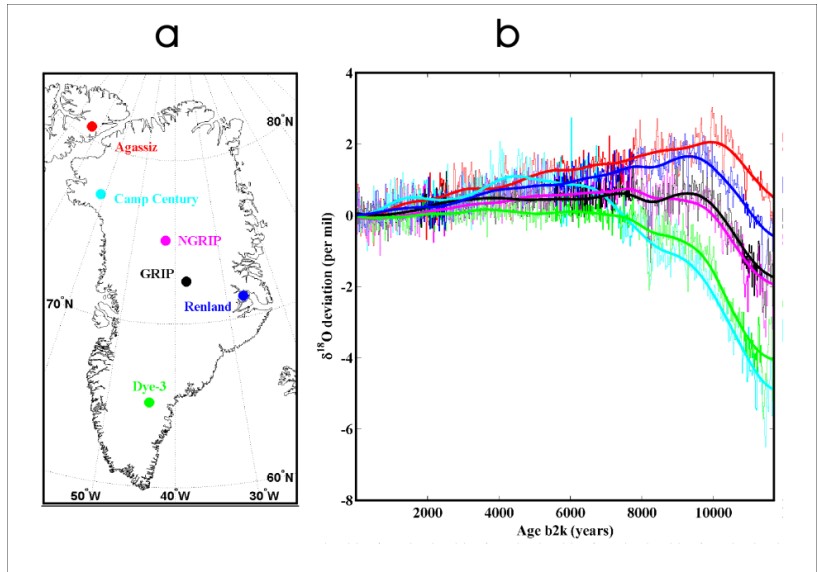


**Figure 4** Ice core sites and  δ($^{18}$O) Holocene records from Greenland and the NE Canadian Arctic
(adapted from Vinther et al., 2009). The elevation corrected average of Renland and Agassiz appears in
Figure 1b (converted to a proxy temperature). Note that the elevation corrections for the Agassiz
records have been recently re-done more accurately (Lecavalier et al., 2017). The argument  presented
herein is not altered by these improvements, however, in that the early Holocene is still the warmest
post elevation correction. Note that all the δ($^{18}$O)  records have been shifted so they are displayed as
deviations from their recent average value.

**4.  The 4.2 ka Event is Expressed in Greenland and Canadian Ice Cores as a Warm Event . How That**
**Signature Might Connect to Lower Latitude Expressions of the 4.2 Event.**
Staying with the Greenland Ice Sheet and Canadian NE Arctic δ($^{18}$O), and melt  records, one can look at
the subject of this conference for another example.  If one takes many  δ($^{18}$O) and melt records from
these ice cores and removes the long term trends and then normalizes all the residuals and makes
various stacked series,  there is a distinct and significant +ve excursion of  proxy temperature at 4.2 ka.
Figure 5 shows this for the δ($^{18}$O) stacks and the single Agassiz melt layer stack, the red vertical shading.
So the 4.2 ka event does show up in the high Eastern Arctic, but as a warm event.





Now I will introduce a hypothesis for connecting the ice core derived proxy climate records from Fig 5 to
the strength of El Niňo and further to the $\delta(^{18}O)$ record from Mt Logan. This hypothesis has been
introduced before (Fisher, 2011) and is meant here purely as an illustration of the likely complexity in
the relationship between various regional expressions of the 4.2 ka event.
Since the North Atlantic ocean provides much of the sinking water  that eventually ends up as deep
water in the Eastern Tropical Pacific  and since the bulk average transport time from the Atlantic to
Pacific  is ~ 1200 years and since El Niňo is thought to be  driven mainly by the difference between
surface and deep water temperatures in the Tropical East Pacific ( $SST_{surface}-T_{deep}$), it has been proposed
that the North Atlantic sector proxy temperature difference [T(t) – T(t-1200a)] is a proxy for the strength
of El Niňo at time t years, (Fisher 2011). It has also been proposed that the 4.2 ka event was triggered by
a period of very strong El Niňos, (Fisher, 2011).
For the sake of continuing this argument, further note that the Mt Logan $\delta(^{18}O)$ has been interpreted as
a proxy for the strength of El Niňo during the Holocene (Fisher et al.,2008; Fisher 2011). Then the Logan
series should correlate with some lagged difference series of the cores in Fig. 5. Figure 6 shows the
correlation coefficient between Logan and the lagged difference series of the Agassiz melt layer series
(solid black line) and similarly between Logan and the 6-core stack of $\delta(^{18}O)$ series (dashed line). There is
a strong correlation when the lag time is ~ 1200 years, which is the mass weighted mean travel time it
takes for subsiding N Atlantic water to end up in the depths of the Eastern Tropical Pacific. Using the 20
year average melt series, the correlation is 0.31, which is significant at the 99.8% level.
Recall that this 3[rd] example is not being presented as correct, (in spite of the correlation significance),
but to show that over large distances and long times in the Holocene the lagged effects of ocean
transport and the resulting juxtapositions in depth of waters of different origins and "ages" should be
considered when correlating records of Holocene length from widely spaced regions. This is especially so
for correlating changes in global moisture flux to changes in the climate of distant polar sites, that may
be initiating the changes at lower latitudes. Simple time correlation of many sites is not likely going to
capture the texture or attribution of the 4.2 event.


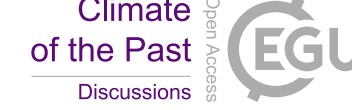

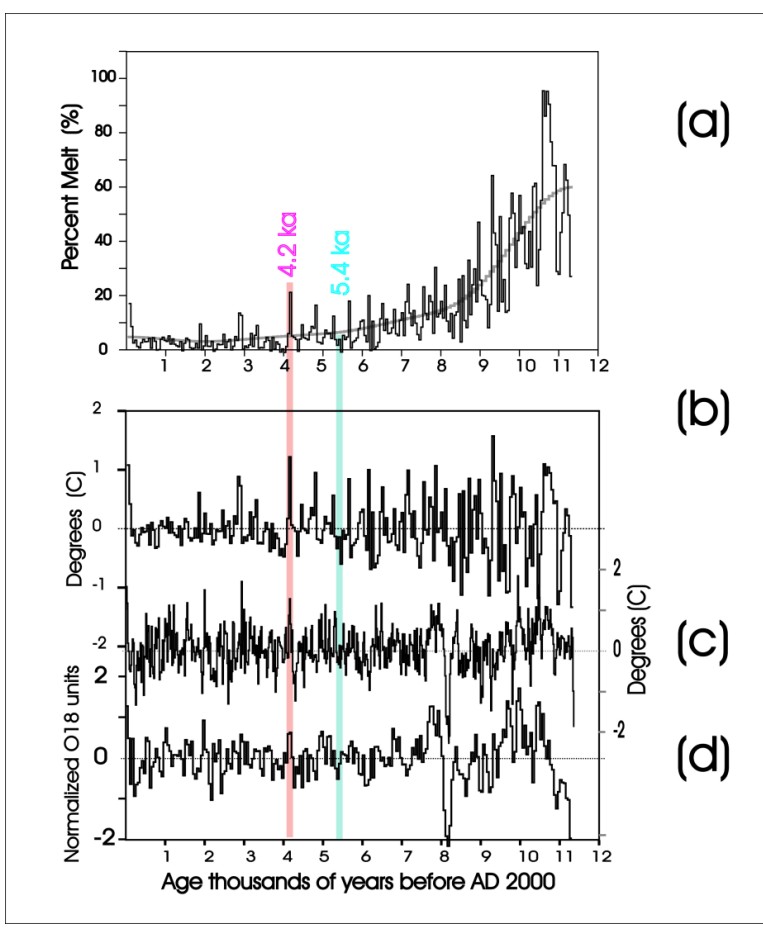

**Figure 5** a) Shows the melt layer series from the Agassiz Ice Cap (Fisher et al., 2012). b) Shows the
residuals of the melt layer series from a 3000 year low pass filter run through the melt series above,
after it was converted to temperature. c) Shows the Agassiz and Renland $\delta(^{18}O)$ stack ,(Vinther et al.,
2009), residuals converted to temperature (see fig 1c . d) Shows the stack made from the normalized
residuals from the Agassiz , Renland and 3 deep Greenland cores after the trends in each series were
removed with a 3000 year low pass filter. The 4.2 ka "event" hi-lited in red is a "warm" event. The
common "cold" event at 5.4 ka is hi-lited in blue.



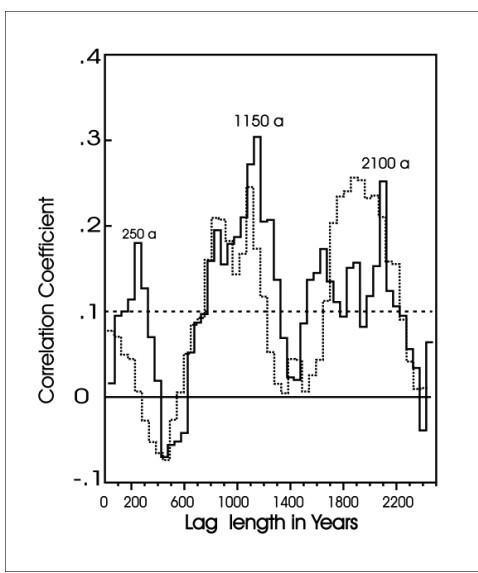

143

**Figure 6** Shows the correlation coefficient between the Mt Logan δ($^{18}$O) series (fig 1a) and the lagged
difference series [X(t)-X(t-lag)] when X is  the Agassiz melt layer residual  series (solid line) and when X is
the δ($^{18}$O)  6-core stack of residuals.  There is a strong correlation coefficient maximum (0.31 using the
melt series)  for a time lag of 1150 years, see (Fisher, 2011).

### Conclusion

So these examples show that for interpretation of sites near the ice sheets and/or variables that are
elevation sensitive and effects that are connected to ocean or uplift related time lags, the  elevation and
time lagged responses are critical in correlating and understanding the proxy records.  Neglecting to
take such account of elevation, its changes with time and lagged time responses can result in erroneous
conclusions.

### Code availability

There is no new code used in this paper.

### Data availability

There is no new data introduced here and uses only previously published data.

### Sample availability

Not applicable

### Author contribution



DAF did it all.
**Competing interests**
There are none.
**Acknowledgements**
Thanks to Dr. Denis Lacelle for help in covering publication fees.

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
