# Peer review of "Holocene Proxy Climate Series Should Account for the Site's Elevation, the Variable's Sensitivity to"

_Climate of the Past, 2018_

## Referee Comment (RC1) · Anonymous Referee #1 · 11 Jun 2018

Introduction This paper makes an attempt to bring arguments for an in-depth consideration of any site's position (elevation, elevation history, distance from ocean) when reconstructing past climates using proxies from the respective site. The first chapters discuss these issues in general, and in the final one, the 4.2 ka event as seen in the Greenland and Canadian ice cores is briefly examined. Both the title and abstract are promising a great debate, however, the main text fails to rise to the expectations – at some points, it's building on fallacies of the same order as the ones that are being discussed (see the detailed comments below). Further, the relevance of the paper for the

"4.2 ka BP event" special issue is debatable, as it only marginally touches on this and the results are nothing new – the two figures are reused from one of the he author's previous articles (Fisher, 2011, figs. 6 and 7b). While the hypothesis of climatic events "seen" in NW Canada are triggered by changes happening in the ocean ∼1200 years before is intriguing and worthwhile considering, it is nothing of new – it has been presented (and better explained) in Fisher (2011). The paper would have befitted from an expansion of the 2011 article, by discussing in more detail the mechanisms, implications and responses on a much wider area. And, as a side note, the text of this final chapter reads in places like conference notes, rather than an article.

Detailed comments Chpater 1 Introduction The introduction could do with a stronger discussion of why the issues raised by the author could be problematic. For instance, except for a few very special cases (e.g. Greenland Ice Sheet), little places experienced a dramatic change in elevation throughout the Holocene that could have affected the stable isotope composition of precipitation feeding sedimentary archives that could provide climate proxies. I would expect such changes to act on longer time-scales (100,000s of years).

Chapter 2 While elevation is important when studying stable isotopes as proxies of past climate changes, it is equally (or more, actually) important to consider not the same isotopes (oxygen, in this case) but the same climatic variable. If the same isotopes are to be considered, it is important that they are measured in the same type of archive – here, two records are from ice cores, and two from lakes. Attempting to compare the absolute magnitude of change is wrong – except for (a limited) diffusion-induced fractionation, the ice cores preserve the original stable isotope composition of snow (e.g. precipitation), while the d18O in authigenic lake calcite records both the stable isotope composition of lake water (and hence of precipitation and post-precipitation processes) and lake water temperature. As such, the original d18O in precipitation, while preserved to a certain extent in the ice cores, it is not preserved in lake d18O. More so, the water of Marcella Lake (studied in this article) undergoes strong evapo-

ration, and as such, the stable isotope composition of water (and further of calcite) is not an accurate reflection of the d18O in precipitation. While the idea of comparing d18O on an elevation gradient is correct (and could be important for palaeoclimatic studies), the way it is done here neglects the post-depositional processes and hence the interpretations are meaningless. I suggest a different approach: rather than using the absolute values, the author could use the relative changes against the long-term mean, i.e., calculate percentages of changes. Perhaps a 5 ‰ change at 5000 m asl, based on ice core d18O is larger than a 1 ‰ change at 1500 m asl, based on lake calcite d18O, but both represent a 10 % change on the Holocene scale – i.e., they are equal. Perhaps. Or maybe not, but it would be more meaningful.

Chapter 3 This issue has been discussed in detail by Vinther et al (2008) who proposed corrections and reconstructed the history of GIS elevation, I don't see what this chapter brings new. Again, it seems to be the expansion of some presentation notes.

Chapter 3 See my comments in the introductory part of the paper.

Conclusions Overall, while some of the hypotheses in this paper are worthwhile discussing, the paper fails to do so. The introduction should be more detailed, and the case should be made stronger by bringing examples where not considering the issues discussed here let to wrong interpretations. The discussion of the d18O change across elevation should consider the type of the sedimentary archive and the climatic variable reconstructed, as well as the syn- and post-depositional history of O isotopes. Part 3 is just an overview of the issues addressed by Vinther et al (2008) and the final discussion on the 4.2 ka event is reloaded from fisher (2011). As such, I cannot recommend publication of this paper.

---

## Referee Comment (RC2) · Anonymous Referee #2 · 24 Jun 2018

Fisher is presenting a short paper regarding the importance of taking into account site elevation, changes in elevation and other local effects on isotopic records before using them in multi-proxy climate reconstructions over the Holocene or longer periods. Three examples of these local effects are presented, mainly focusing on the Arctic area. Although in a moment where a lot of regional, continental and global paleoclimate reconstructions are attempted, a paper dealing with this argument would be desirable and of high value, the paper fails in its objectives, mainly for not having a paper structure and for not presenting any new data. The paper is presented in the form of a conference

presentation. Would be the paper be a review? Would the paper be focusing on the 4.2 ka event? Other events are presented apart from the 4.2 ka. No new data are apparently presented and a general context is lacking in the introduction, making the readability difficult. Some more specific comments: Page 1, line 15: the introduction must be enlarged and should be well structured. No reference to previous papers on these arguments are presented. A general context is lacking. It should be re-written. Page 1, line 16: what it is reported in the manuscript is not only elevation. Please, add. Page 1, line 23: The title of the paragraph is too long. I would suggest rephrasing into: Stable isotope signatures of the same event at one geographical location. Page 1, line 25: please, take away the parentheses before 18 and after O or explain the reason for having them. Please correct in all the manuscript. Page 1, line 29: "nearby sites" …. First a map could be useful, then it should be better specified that here we are dealing with different climate archives and climate proxies having different interpretation and calibrations against present day climate. Page 2, line 34: which is the model? To which simulation you are referring? Which type of simulation is it? Page 2, line 36: after stable isotopes: at which archive you are referring? Page 2, line 42: "as does its nearest …": no explanation is reported, probably it is in the original paper. Page 4, line 68: also in this case the title of the paragraph is too long. Please, change it. Page 4, lines 70-80: it seems an abstract of the paper by Vinther et al., 2009… Page 5, line 82: has caused some investigators: please add references. Page 5, line 96: paragraph title to be changed. Page 5, line 98: "… subject of this conference…": please change this sentence. Page 5, line 100: +ve: should this means positive? Please modufy. Page 5, line 101: modify "stack" into record. Page 5, line 101: ….. the red vertical shading: what?? Something is missing. Page 6, lines 103-105: If this has already been published, what it is new in the hypothesis reported below? Page 6, lines 107-113: please add some references here, apart from Fisher et al. Figure and figure captions Figure 1a: this figure should be enlarged: the 1835 AD event is not visible at all. Figures 1b and c are not discussed in the text, only at the end. Figure 2 caption, line55: I would change the word "records" into archives. Line 57: "model": which model? Figure 5

caption: line 137: hi-lited: please modify. Line 138: add a reference after "common cold event at 5.4 ka". This 5.4 ka event is not clear at all in the figure. There are a lot of other similar peaks . . ... what makes this peak interesting against the others? Figure 6: Please add a legend to the figure. I would not recommend the publication of the paper in its present form but only a new resubmission after considering all the comments above, structuring the manuscript as a paper, clearly focusing on the 4.2 ka event (if this is a special issue on this), adding a real introduction and possibly presenting new data or at least considering to make a more robust review of the arguments presented.

---

## Author Comment (AC1) · 15 Aug 2018

Response to Editor of "Climate of the Past" and reviewers of "Holocene proxy climate series should account for the site's elevation, the variable's sensitivity to elevation history and time-lagged effects: three examples ": CC 2018 55 The ms was submitted as a discussion piece for the special issue "The 4.2ka BP Climatic Event" First, I thank the reviewers taking time to do their work. They both thought the ms did not contain anything new that had not been covered in some of my earlier papers. They are correct. But, I did not think that the discussion here needed to have new work presented.

[Figure]

I have felt that the paleo -reconstruction community have implicitly assumed that site elevation and history need not be taken into account as a first order variable. Presenting three examples from the literature that show this assumption is false was the goal of the ms. I probably should have called them counterexamples that disprove the implicit constant-elevation-assumption. The 'constant elevation implicit assumption' is still used even over Holocene-long intervals, (eg. Briner et al.,2016) so I felt the discussion of its' validity is relevant. Possibly the "Climate of the Past" was not the right outlet for the proceedings of a small conference. When confronted with the dilemma presented by the reviews, I made repeated and failed attempts to get in touch with a human in the editorial process for guidance. Now, I have too many commitments to meet the journals' deadlines. In short please take my ms out of any further consideration for publishing. D Fisher Aug 14 2018